# From Name to Myth (Based on Russian Cultural and Literary Tradition)

Olesia D. Surikova * and Elena L. Berezovich *

Department of Russian Language, General Linguistics and Verbal Communication, Ural Federal University, Yekaterinburg 620075, Russia

*   Correspondence: surok62@mail.ru (O.D.S.); berezovich@yandex.ru (E.L.B.)

**Abstract:** This paper analyzes the cases wherein a previously unknown and unique mythological character (with his/her specific behavior, "personal" traits, appearance, origin, etc.) is generated by a cultural linguistic sign or a fragment of text. This research is based on the Russian cultural and linguistic tradition, mainly in its dialectal version (the language of Russian peasants). Its sources include data published in the late 19th–early 21st century in dictionaries of Russian dialects and, primarily, the unpublished field materials of the Ural Federal University Toponymic Expedition, covering data from the Russian North, the Urals, and the Volga region. According to their nature or origin, the names of characters studied in this paper derive from two types of linguistic signs: (1) Names based on usual forms of standard vocabulary that can be both proper and common nouns; the former may refer to various categories, such as toponyms (names of geographical objects), chrononyms (names of calendar dates), hagionyms (names of saints), names of icons, etc. (2) Names originating from a text, usually folkloric; these are word combinations or phrases that only act as a single unit within their "parent" text. Sometimes, but less often, these consist of one word that is of key importance in the source text. Such a phrase or word can migrate outside the "parent" text or genre, expanding their lexical combinability and changing their syntactic regime to become a name of a mythological character. It takes two sources of motivation for a new character to emerge—a linguistic (a word that seeks a new context) and a cultural one (a semiotically intense context, such as a situation associated with danger, prohibition, omens, aggression, or magical practices). The combination of these incentives is not uncommon, so the stock of mythology used for names is being constantly renewed.

**Keywords:** Russian linguistic and cultural tradition; onomastics; mythology; folk Christianity; folklore; ethnolinguistics; name of a mythological character

## 1. Introduction

The folk religious–mythological system is a constantly evolving and developing paradigm. This development is largely driven by the resources of the natural language used by the community. However, in the public mind, mythology and folk religion are often regarded as static sets of values and principles formed sometime in the (very distant) past and sanctified by its authority. Advocates of "folk antiquity" deny the whole idea of changing the religious–mythological system, and such mythological purism is not unfounded. To recognize the predominant influence of folk religion and mythology on the cultural and linguistic worldview, social practices, and psychology, the naive consciousness must assign it with unquestionable authority. This attitude to mythology is mythological itself, fitting into the *Golden Age–Iron Age* opposition that idealizes the past and denounces the present.

In this article, we would like to demonstrate once again that the religious and mythological system is not a rigid framework but rather a living organism that is constantly developing and replenished with new characters and ideas (or at least shades of ideas and

variants of characters). We will not cover the cultural mechanics of drawing in modern realities or figures (mostly political), as this is well described by folklorists and ethnographers. For example, a vast corpus of texts about Lenin and Stalin appears in 20th century Soviet folklore, while the tsars—Peter the Great or Catherine the Great—assume the same position in Russian folklore recorded in the 19th century. Here, the authors will show that the folk religious–mythological system can be powered by internal resources, using only old (more or less archaic) motifs, plot connections, and character types, and demonstrate how new folkloric–ethnographic facts emerge on this basis. (In this case, the strict opposition between "old" and "new" folklore, on which many researchers insist, is invalid. "New" phenomena can appear within the old system of ideas that are actually indistinguishable from the "old" cultural facts, unless researchers undertake special verification procedures. This means that for the folk tradition, innovative phenomena have no less weight and significance than old phenomena and are not essentially different from the latter, since they represent that same system of ideas).

## 2. Research Problem and Materials

It is obvious that it takes a certain stimulus for this new phenomena to emerge from old religious and mythological ideas. We will consider a set of cases wherein this stimulus would be an inspiration from a linguistic sign. This research is broadly based on the Russian cultural and linguistic tradition, mainly in its dialectal version (the language of Russian peasants). Its sources include data published in Russian dialect dictionaries (second half of the 19th–early 21st century) and, primarily, the field materials of the Ural Federal University Toponymic Expedition in the Russian North, the Urals, and the Volga region. The authors have been engaged in this fieldwork for many years, and the materials used in this paper were recorded in the late 20th–early 21st century. Additionally, this paper also uses some examples from the 20th century book tradition (though socio-linguistically marked as urban or "high" language) to substantiate the productivity of the cultural and linguistic mechanism under discussion. In addition, the authors draw parallels from foreign linguistic and cultural traditions.

The studied pattern consists of the following: a unique character (with his/her specific behavior, "personal" traits, appearance, origin, etc.) that never existed before (never appeared as such or as a version of another known character) emerges into the religious and mythological system, brought by a cultural linguistic sign or a text fragment.

The mechanics of mythmaking builds on the perception of some phenomenon or a situation as irrational, supernatural, or implying danger, restriction, bad luck, communicative aggression, and so on. These ideas become articulated in certain names or texts (for instance, a curse) that are characterized by "semiotic tension", correlating with the tension of the experienced situation.

Characteristically, this tension can be verbalized in different forms, sometimes unrelated to the character. These may be words or word sequences used to denote a geographical site (e.g., a river or forest), an object (e.g., an icon), or a phenomenon (e.g., a disease), and so on. But the tension is still seeking a "subjective" resolution: there must be a subject causing the action, such as a character (the one who causes the disease or has the ability to punish, etc.) that acquires a certain name. Thus, being put in the situation of semiotic tension, the initial word or phrase is reinterpreted and placed in a new syntactic context, in which it designates a character, an acting subject. At once, this new creature is given a name (i.e., it needs to be named to become a creature) and acquires flesh and blood: its name spreads out into other texts, including works of a different genre (different semiotic context), the lexical combinability of this name broadens, and the new character acquires new features based on phonetic attractions or cultural allusions (however, the key feature—which may remain the only unanimous one—is the character's ability to act). This is how the linguistic fact as such goes beyond the limits of language and acquires a "superstructure" of collective ideas that moves it to the realm of mythology.

By their nature or origin, the names of characters studied in this paper derive from two types of verbal signs: (1) "Lexemic" names based on usual forms of standard words in the given language that can be both proper and common nouns; (2) "Textual" names originating from a text, usually folkloric.

Let us examine both types consecutively.

### 3. Analysis

*3.1. Names of Characters Derived from Standard Units of the Lexical System*

Both proper and common nouns can be used as derivational bases. The former can refer to different classes: toponyms (names of geographical objects), chrononyms (names of calendar dates), hagionyms (names of saints), etc.

*3.2. Names of Characters Formed from Proper Nouns*

3.2.1. Toponym → Name of a Mythological Character

The images of mythological creatures can be based on the names of geographical objects, i.e., toponyms. There are repeatedly attested cases wherein the name of a *genius loci* is derived from a toponym. The tradition of naming mythological characters according to their habitat is particularly strong in the Russian North (see Cherepanova 1983). Typically, this would refer to dangerous places, remote from the settlements and perceived as scary, "unclean", or "alien". The character associated with such a locus is invented in order to warn people of the danger and tacitly forbid them from visiting this place. This is what creates the "semiotic tension" that is essential for stimulating the potential of the noun to be a name.

*Case 1.* Near the village of Doman, (Makarievsky district, of the Kostroma region), there is a small river, *Заря́нка* (*Zarianka*), flowing into the lake Sosnovskoye. According to the locals, the name of the river is linked to the fact that it flows from west to east, i.e., towards the dawn, cf. Rus. *заря́* (*zaria*) 'dawn' (TE). At the same time, local old-timers say that people used to associate this toponym with the image of a mermaid named *Заря́нка* **(Zarianka),** who sits on a cape at the point where the river flows into the lake, and brushes her hair: *Рано утром сидит, в четыре утра. Не ходи, говорили, на озеро: Зарянка утащит!* 'She sits there early in the morning, at four. Don't go to the lake, they said, Zarianka will drag you away!'. Thus, *Zarianka* was supposed to scare children and prevent them from going to the water alone; the name of the character that is capable of action (who can drag a person away) is formed from the name of the river.

*Case 2.* Let us consider a group of cases wherein toponyms that "come to life" denote remote forests or swamps which are frightening or potentially dangerous (as one can become lost, disappear, or drown there). Residents of the Velsky district (Arkhangelsk region) believe that the *Пáтрома* (*Pátroma*) bog is inhabited by a character named *Пáтромиха* **(Pátromikha)**, who is also sometimes referred to as a "white woman", cf. Rus. *бéлая бáба* (*bélaia bába*) 'white woman' (TE). Peasants in the Verkhnetoyemsky district (Arkhangelsk region) believe that in the *Чу́дницы* (*Chiúdnitsy*) forest, there lives a creature named *Чу́дница* **(Chiúdnitsa)** (TE). The forest name is formed from the Arkhangelsk dialect word *чу́дница* (*chiúdnitsa*), meaning 'hunting trail; ski track' (DRND). Later on, this name was semantically attracted to the standard Russian word *чу́до* (*chiúdo*), meaning 'miracle; supernatural phenomenon; extremely unlikely event', which contributed to the emergence of this mythonym. Residents of the Verkhovazhsky District (Vologda region) think that the forest river *Кóленьга* (*Kólen'ga*) contains a mythical mistress named *Кóленьга* **(Kólen'ga)** (TE). Inhabitants of the Griazovetsky district (Vologda region) tell stories about scary creatures named *Ухан ы́* **(Ukhaný)** from the *Уханíха* (*Ukhaníkha*) forest (TE). In the latter case, the connection to the verb *у́ хать* (*úkhat'*) meaning 'hoot; to make a loud, sharp sound; to shriek loudly and abruptly' is important. This verb denotes, among other things, the cry of an eagle owl, and the sound effect of an echo (which is a frightening phenomenon in the forest), as well as sounds made by evil forces. A similar fact, based on the fear of loud sounds and of the forest echo, was recorded in the Nandomsky district (Arkhangelsk re-

gion). In connection with the name of the *Вóешный* (*Vóeshnyi*)[1] brook, the informants note: *На Вóешном Ручье **Вóюшко** воет* 'On the Vóyeshnyi brook **Vóiushko** howls' (TE). The people of the Onega District (Arkhangelsk region) believe that in the *Хéмерово* (*Khémerovo*) forest, there lives a creature with the surname **Хéмеровский (Khémerovskii)**; this character leads off the road those who come to the forest (TE).

*Case 3.* Near the town of Polevskoi, in the Sverdlovsk region (Middle Urals), there is the *Azóv* (*Азóв*) Mountain, known for the numerous treasures it allegedly hides. The mountain is associated with its mythical "mistress", *девка Азóвка* (*maiden Azóvka*), whose image appears in numerous legends (Krugliashova 1991, pp. 69–72; VTES n.d.). The name of this character has been an object of verbal magic. Explaining where Azovka might come from, residents of Polevskoi mention a story of a curse, according to which the parents of the heroine (who used to be an ordinary factory girl) cursed her for connecting her life with robbers (Krugliashova 1991, p. 70). In the folkloric ideas surrounding Azovka, the curse is more than a fact of the past; it has transformed into various meta-language motifs. For us, the primary one is that the "actionable" name of the character can either open treasures or destroy those who come for them. The treasures can be opened to those who guess Azovka's cursed name and speak it (Ibid., p. 69). A version of this legend was used by the famous writer Pavel P. Bazhov in his tale *A Dear Name*. In this tale, a girl (with an unknown name), living in the Azov mountain, suffers her lover's death. As he dies, he reveals to her that one day, another young man will come to the mountain and loudly call out the heroine's name. Then, she must come out of the mountain and go to him, giving the hidden treasure to the people (Bazhov 2019, p. 21). Apparently, Bazhov omitted the heroine's name so that the motive of having to guess it would look more plausible (this name serves as a magical key to the treasure).

There are other versions of this story with more complex meta-linguistic contexts. According to one of these variants, a cursed queen moans in the mountain. If her name matches the one of a stranger who approaches the mountain, she will stop moaning because her curse will disappear (Blazhes 1983, p. 8). According to a second version, recorded in the village of Krylatovsky (Sverdlovsk region), one should keep silent while going up Mount Azov, as if somebody calls their companion by his/her name and the name coincides with that of the "mountain spirit", the spirit will take the owner of the name (VTES n.d.).

Presumably, the name *Азов/ка* (*Azov/ka*) has been reconceptualized[2] because of its connection to the words *зов* (*zov*), 'a call', and *звать* (*zvat'*), 'to call'. Such a connection is caused by the "magic of calling", which is a term for invoking a person's name in the traditional Slavic culture determined by "the mythological identification of a person and his/her name and, at the same time, by the exceptionally mediative properties of the human voice" (Agapkina 1999, p. 350). Thus, the mythonym here is generated by the toponym due to the link between *Azov* and *zov*, meaning 'call', with reference to the folkloric beliefs around invoking one's name.

### 3.2.2. Chrononym → Name of a Mythological Character

The names of calendar dates (chrononyms) are another source of the names of mythological characters. It so happens that chrononyms in the Russian folk calendar are largely derived from the names of saints (hagionyms), wherein the word-formative derivation can be expressed both explicitly and implicitly. For example, 2 August, the day of the prophet Elijah, can be denoted by the following possessive forms: *Ильúн день* (*Il'ín den'*); *Ильúнский день* (*Il'ínskii den'*), 'Elijah's Day'; and *Ильúн праздник* (*Il'ín prazdnik*), 'Elijah's Feast' (Atroshenko et al. 2015, pp. 184, 188, 190). Other variants include the suffixal noun *Ильúнщина* (*Il'ínshchina*) (< *Ilia* 'Elijah') (Ibid., p. 190), or a word that formally coincides with the name of the saint, such as **Илья́** (*Iliá*), 'Elijah', and **Илья́-Проро́к** (*Iliá-Proŕok*), 'Elijah the Prophet' (Ibid., pp. 190, 194). The latter case especially emphasizes the fact that the character's name is initially embedded into the chrononym.

Speaking about characteristic features of the traditional Russian worldview, Svetlana M. Tolstaia notes "the anthropomorphic nature of calendar time, the personification of

days (holidays, feasts), the tradition to identify the day and its mythological patron, as well as the demonology of personified holidays" (Tolstaia 2005, p. 383). Thus, if there is a thunderstorm on 2 August, people would say, *Илья-Пророк на тучах катается*, meaning, 'Elijah the Prophet rides on clouds' (Kostroma region) (Atroshenko et al. 2015, p. 195). Similarly, the idea that after 2 August, the water in rivers and lakes becomes too cold and unsuitable for bathing has become formalized in such idiomatic expressions as *Илья-Пророк льдинку в воду бросил*, meaning, 'Elijah the Prophet threw a piece of ice into the water' (Vologda region) (Ibid., p. 195). At the same time, the "personified day" is not equal to the saint character it is named after. The semantics of these chrononyms always include a complex of the beliefs and tokens associated with the calendar date itself, and ideas about the ritual actions performed on this day (rather than facts about the saint described in the Apocrypha, or hagiographies). Therefore, naturally, *Ilia the Prophet* who puts a piece of ice into the water of a river near Vologda on 2 August is not identical to the biblical prophet Elijah, who lived under King Ahab. But this is not yet a new mythological character. Such new characters can arise on the basis of restrictions and bans associated with these calendar dates, in prohibitive formulas, proverbs, folk stories, beliefs, etc. In this case, the image of a new mythological character will embody the consequences of violating the ban on any actions carried out by a person on a given calendar date; the character's actions will be connected with the punishment of a person for violations of the ban. The character will hence become a kind of "namesake" of the saint in whose honor the calendar date is named. Let us consider how this occurs.

*Case 1.* The name of the Great Martyr Barbara of Iliopolis (Nicomedia) is the basis of the chrononym *Варварин день* (*Varvarin den'*), 'Barbara's Day', celebrated on 17 December (Atroshenko et al. 2015, p. 60). It is believed that on this day, severe frost and ice arrives, the sleigh track is established, and the daylight hours start to increase. These ideas are reflected in expressions in which the chrononym coincides with the hagionym in its form, and essentially becomes a "personified holiday". Some examples are: *Варвара мостит*, 'Varvara/Barbara is paving' (regarding the appearance of ice, Kama region), and *Варвара ночи украла, дня притачала*, 'Varvara/Barbara stole the night and added the day' (regarding the lengthening of the daylight hours, Nizhny Novgorod region) (Ibid.). Apart from meteorological tokens, Varvara's (Barbara's) day is also associated with some socio-cultural prohibitions; for example, on 17 December and the following days, women were forbidden to spin flax or wool. This prohibition was known not only to Russians, but also to other Slavs. So, the Belarusians in the Brest region also believed that it was forbidden to spin on this day, *бо вона вэртёнами замучэна*, 'because she was tortured by spindles' (Tolstaia 2005, p. 42). This prohibition gives birth to a new mythological character, also associated with Varvara's day but not rigidly attracted to this date. In the suburbs of Moscow, they tell stories about a creature named **Varvarka**, who appears if you spin flax or wool on days when it is forbidden: *Ежели прясть на Святки, так Варварка придёт*, 'If you spin on Sviatki (the Twelve days of Christmas), Varvarka will come'; *Ежели что затеешь в праздник, тётка моя всегда ругается: "Грех <…> Смотри, Варварка придёт"*, 'If you start something on a holiday, my aunt always scolds: It's a sin <…> You watch, Varvarka will come!' (Atroshenko et al. 2015, p. 61). These contexts do not specify what will happen to the violator of the prohibition if *Varvarka* comes to them, but, obviously, the appearance of this character should cause fear. *Varvarka* resembles the Slavic *kikimora* (*kikimora* is a Russian and Belarusian mythological female character living in a human home, harmful to the household and its people), who has a similar role (punishing for violations of the ban on spinning) and is clearly not friendly[3]

*Case 2.* On September 11, the Orthodox Church celebrates the Day of the Beheading of John the Baptist (Atroshenko et al. 2015, pp. 174–75). This date is associated with the prohibition to pick and eat any round vegetables (cabbage, potatoes, turnip, etc.) because their shape resembles the severed head of the Baptist. Let us note that the real motivation for this prohibition is the need for time to stock up on vegetables for the winter before this date, which is caused by weather conditions. In addition, since the Day of the Beheading

of John the Baptist is a bloody and tragic date, a strict fast begins a week before September 11, which implies the abstinence of meat, fish, and red berries (resembling blood) (Ibid., p. 174). In this way, this date receives a special name in the folk calendar—*Ивáн-Пост* (*Iván-Post*), 'Ivan/John the Fast'; *Ивáн-Постúтель* (*Iván-Postítel'*), 'Ivan/John the Fasting'; or *Ивáн Пóстный* (*Iván Póstnyi*), 'Ivan/John the Lenten' (Ibid.). This folk chrononym and the prohibition to work in the vegetable garden on September 11 created a separate mythological creature called **Ивáн Пóстной (*Iván Póstnoi*)**, 'Ivan/John the Lenten', known in the Middle Urals. This character is a vegetable garden spirit that punishes people who come there at the forbidden time (Matveev 1996, p. 207). Residents of the Arkhangelsk region also knew about Ivan the Lenten; they forbade children to go to the vegetable garden, scaring them by saying that he could cut off their heads (Moroz 2007, p. 65). It is interesting that ideas about Ivan the Lenten go beyond verbal prescriptions. In folk culture, he can be embodied as a character of the autumn folk play: *В огороде-то Иван Постной, не ходите. Ребятишек пугать---какая-нибудь старушка нарядится, шубу навыворот наденет, вот и Иван Постной*, 'Ivan the Lenten is in the vegetable garden, don't go there. It's for scaring children—some old woman will dress up and put on a fur coat inside out, and here is Ivan the Lenten' (Matveev 1996, p. 207). In the Middle Urals, there is a synonymous character named **Ивáн Капýстник (*Iván Kapústnik*)** (< *капуста* (*kapústa*) 'cabbage') (Ibid., p. 226). This is also another scary autumn spirit whose name refers, on the one hand, to the main dish of the fast—cabbage—and on the other hand, to the prohibition of picking and eating round and large (head-like) cabbage on September 11.

*Case 3.* There are cases in which a chrononym "comes to life" not only as a mythological character endowed with certain characteristics in the texts of popular culture, but also as a character of a guisers' play (cf. the testimony regarding Ivan the Lenten above). Thus, the Thursday before Trinity in Russia is universally known as **Семúк (*Semík*)**. In the Nizhny Novgorod region, the word *Semík* and its feminine derivative *Семичúха* (*Semichíkha*) are the names of the male and female characters of the guisers' play on this day: *В Семик <…> кто-либо из девушек наряжался Семичихой. Рядили Семика (парень наряжался стариком) и Семичиху (девушка наряжалась в старуху)*, 'On Semik <…> one of the girls dressed up as Semichikha. They dressed up Semik (a boy disguised as an old man) and Semichikha (a girl disguised as an old woman)'; *Ходили по домам собирать продукты для яичницы, водили Семика и Семичиху (женщина переодевалась в мужчину, мужчина—в женщину)*, 'They went door to door to collect products for cooking fried eggs, they took with them Semik and Semichikha (a woman disguised as a man, a man dressed up as a woman)' (Atroshenko et al. 2015, p. 392).

An interesting fact confirming the persistence of this pattern is described in an article by Vladimir V. Napol'skikh (2019, p. 142). This article focuses on the traditional rituals of the Krasnoufimsk Udmurts (Votiaks), an isolated pagan group within the Finno-Ugric people who live in the Middle Urals. Through the Mari (also pagan) mediation, the Votiaks borrowed the features of the Orthodox Russians' Sviatki rituals. Like the Russians, on 13 January the Kransnoufimsk Udmurts celebrate the middle of Sviatki, which is a calendar period (6–18 January) characterized by folk plays, fortune-telling, and ritual outrages. In the folk calendar of Orthodox Russians, 14 January is called *Васúльев день* (*Vasíl'ev den'*), meaning 'Vasilii's/Basil's Day', or *Васúлий Велúкий* (*Vasílii Velíkii*), meaning 'Vasilii/Basil the Great: the day of St. Basil the Great', and the evening before this holiday (13 January) is called *Васúльев вечер* (*Vasíl'ev vecher*), meaning 'Vasilii's/Basil's Evening'. In the Mari language, the name for the evening of 13 January is a half-calque: *Basil kuɣuza*, 'Vasilii/Basil the Lord' (Mari *Basil* < Rus. *Василий*). In the language of Krasnoufimsk Udmurts, this name is a borrowing from Mari: *Baśilʲ kuɣuźa*; that said, the pagan Udmurts know nothing about the Christian saint and the Orthodox day of his commemoration. On 13 January, the Krasnoufimsk Udmurts go to the baths, perform a prayer conducted by the priest *molla*, eat the ritual pastry *tabani* (pancakes with butter), and dress up. One of the typical disguise characters is in fact named with the same word as the whole holiday—**βaśilʲ kuɣuźa*, 'Vasilii (Basil) the Lord'. "In the evening a team of ten or more adult married men in masks of

bears, horses, geese, in disheveled clothes, etc., would move from house to house. One of them played the role of the old man *βaśilʲ kuɣuźa*" (Ibid., p. 142). The costume players were led by a character called *βaśilʲ kuɣuźa*, and went to the houses where children were waiting for them. The children then had to demonstrate their labor skills (spinning skills for girls, and the skills of weaving lapti for boys). If the children failed with their tasks, *βaśilʲ kuɣuźa* would frighten them: *nuša koškom tone, ləməje bəćkaltom, kulod!*, 'We'll drag you away, we'll throw you out into the snow, you'll die!'; *ńulešti ləktimə, tone no nuom!* 'We came from the forest and we'll carry you away!' (Ibid.). Once again, we face a situation of semiotic tension formed by the fear of the people in disguise. On the one hand, they portrayed characters of lower demonology, and on the other hand, they were demonized as separate (calendar-related) beings (this is also true for the Northern Russians).

3.2.3. Name of an Icon → Name of a Mythological Character

The worship of Virgin Mary plays a very special role in the Russian Orthodox tradition, together with the calendar holidays, churches, monasteries, and icons dedicated to her. It is not surprising, therefore, that the names of Virgin Mary's icons and the features of their worship have become a source for creating new mythological beings. In this case, the situation of semiotic tension unfolds in a different perspective; here, it would be more accurate to use the term "semiotic intensity". Using the language of dramaturgy, in the Virgin Mary stories, the main plot twist and stimulus is not fear or prohibition, as was described above, but a precedent cultural text, the apocryphal motif of Virgin Mary's travels (*хождение---khozhdenie*), which was widely spread in the folk Christian tradition, or the legends surrounding the appearance of Virgin Mary's icons. In short, this is a later cultural "superstructure" of the Gospel image.

*Case 1.* One of the most common types of Virgin Mary's depiction with the infant Jesus Christ in the Orthodox iconographic tradition is the Hodegetria (< Greek Ὁδηγήτρια 'She who leads the way'), Rus. *Одигúтрия* (*Odigítriia*). An icon of this type is a frontal half-length image of Virgin Mary ("Theotokos" in Eastern Christianity) pointing with her hand to Jesus. This type includes such widely revered icons in Russia as the Tikhvin, Smolensk, Kazan, Georgia, Iviron, Pimenovskaia Theotokos, Troeruchitsa ("three-handed Theotokos"), Passion Theotokos, the Black Madonna of Częstochowa, and others. Many churches and monasteries were consecrated in the name of Virgin Mary Hodegetria. Among them is *Арсениево-Маслянская Одигитриевская мужская пустынь* (*Новая пустынь Пресвятой Богородицы Одигитрии новоявленной, что во мхах*), or 'Arsen'evo-Maslianskaia Hodegetria Male Hermitage' ('New Hermitage of the Most Holy Virgin Mary Hodegetria of the New Appearance, which is in the Marshes'). This monastery is 40 km away from Vologda and, as the legend says, was founded on the spot where a Hodegetria icon miraculously appeared, near the Masliana River. The legend about the appearance of the icon, which was initially connected to this specific location, has since become "blurred" in the popular religious worldview (as often happens). The legend lost its specific toponymic reference, and, in addition, was influenced by the motif of Virgin Mary's travels. As a result, in different villages of the Vologda district (situated within the Vologda region), residents speak of holy places (springs with holy water, stones with healing powers) that received special properties because a female character named ***Яúтра* (*Iaítra*)** passed by them. The word *Яúтра* is a phonetically transformed name of the Hodegetria icon (Rus. *Одигúтрия---Odigítriia*). In some cases, *Iaitra* is identified with Virgin Mary: *Она шла, Яитра, Божья Матерь, так на камешке следочки от ладошек остались, около Святого Колодчика*, 'She was walking, Iaitra, the Mother of God, so on a stone there were traces from the palms of her hands, near the Holy Well' (TE). In other cases, *Iaitra* is viewed as a separate saint: *Яитра была, она святая. Ходила по нашей земле. Где пройдет, там святые камни находили*, 'There was Iaitra, she is a saint. She walked on our land. Where she passed, people found holy stones' (TE).

### 3.3. Names of Characters Formed from Common Nouns

In this case, the names of characters emerge on the basis of their phonetic similarity with a common noun. The names that appear here are often quasi-anthroponyms of a kind. They coincide with personal names only formally, but unlike typical Russian personal names, they have a secondary inner form that is employed in the wordplay.

*Case 1.* In Kostroma dialects, the verbal form *лепить* (*lepit'*), meaning 'to mold, to stick (to), to seal up, etc.', and the adjective *липкий* (*lipkii*), meaning 'sticky', have been combined into the expression *баба Лúпа* (***Baba Lípa***), meaning 'snow woman' (DRND). The element *Липа* (*Lipa*) here coincides with the diminutive form of the name *Olimpiada* (in common parlance—*Lipiada*) ([Petrovskii 1980], p. 213). To make a *Baba Lipa*, three snowballs were placed on top of each other: first a large one, then a smaller one, and then a still smaller one. The upper snowball was then "fashioned" into a human head; the nose was a carrot and pieces of coal served as the eyes, and on the head there was a bucket. The *Baba Lipa* was perceived as a personification of winter, and on Maslenitsa she was destroyed (if she had not herself melted by then). People attributed the action of "molding" or "sealing up" to this character due to its name's phonetic similarity with the verb *лепить* (*lepit'*), meaning 'to mold, to stick (to), to seal up, etc.' This verb then became a contextual neighbor of the quasi-anthroponym *Baba Lipa*, and was thus used in the phrasing of a prohibition, whereby if there was a severe frost outside, children were not allowed to go for a walk and were threatened as follows: *Мороз сильный, нас на улицу не пускали, говорили: "Баба Липа глаз залипит"*, 'The frost was severe, we were not allowed outside, they said: Baba Lipa will cover up (seal up) your eye [with snow]' (DRND). Therefore, to create a mythological character within the described pattern, two basic conditions must be met: (1) the character's name appears in the context of a wordplay; (2) the character's name appears in a semiotically tense context (i.e. in the phrasing of a prohibition).

*Case 2.* This is a case related to the prohibition of late trips to the bathhouse (*banya*) due to the risk of getting carbon monoxide poisoning. This prohibition was motivated by there being some creature capable of harming a person (especially a child, since the phrases used for prohibitions are most often directed at children and have pedagogical functions). In Kostroma dialects, this creature is sometimes referred to with a quasi-anthroponym that is phonetically similar to the meaningful appellative *дым, ды́мно* (*dym*, *dýmno*), meaning 'smoke, smoky', and coincides with a personal name **Дóмна (*Dómna*)**: *Не ходи в баню, туда Домна зашла. Детей пугали, чтоб не шли, когда угар*, 'Don't go to the bathhouse, Domna has come in there. Adults used to scare children so they wouldn't go there when it was smoky' (TE).

*Case 3.* This case is described in the work of [Berezovich and Rut] ([2016], pp. 89–90). In Russian, there is a well-known name for a children's disease, *родúмчик* (*rodímchik*), meaning 'a seizure accompanied by convulsions and loss of consciousness'. This name is related to the verb *родúть* (*rodít'*), 'to give birth', and the noun *рóды* (*ródy*), 'labour'. This name appears in such phrases as *родимчик взял* (*rodimchik vzial*), lit. 'rodimchik has taken [the child]', and *родимчик бьет* (*rodimchik b'et*), lit. 'rodimchik is beating [the child]'. In these phrases, the disease, as often happens, is personified, and the status of being an acting figure is attributed to it. In the Kostroma region, the expressions denoting such a seizure are *Рóдька пришёл* (*Ród'ka prishel*), lit. 'Rod'ka has come', and *Рóдя взял* (*Ródia vzial*), lit. 'Rodia has taken, seized [the child]'. Here, the disease is not named directly, but euphemistically—by means of a phonetically similar quasi-anthroponym ***Рóдька, Рóдя* (*Ród'ka, Ródia*)** (which coincides with the diminutive forms of the male personal name *Родиóн—Rodión*): *Ребёнок капризничает сильно, катается даже, бьётся на полу---дак Родька пришёл, Родион---имя, Родька*, 'A child is very capricious, even rolls around, beats on the floor—that means Rod'ka has come, Rodion is the name, Rod'ka' (DRND). *Рóдька* escapes the limits of the "maternal" phraseological unit and enters the phrases that parents use to scare their children (i.e., into a semiotically tense context). In such texts, the combinability of this name expands: *Ну, реви-реви, за углом Родька стоит, придёт, тебя заберёт*, 'Well, cry-cry, Rod'ka is standing around the corner, he will come and take you away'; *Родька придёт и заберёт*

*тебя. Видно, страшилище было*, 'Rod'ka will come and take you away. He must have been a bogeyman'.

## 4. Text-Based Character Names

These names come from a text—as a rule, a folkloric text. Initially, they have nothing to do with proper names, as these are word combinations or phrases that can only act as a single unit within their "parent" text. Occasionally, this name is a single word that is of key importance in the source text. This phrase or a word can migrate outside its initial text or genre (often due to its expressiveness, frequent recurrence, etc.), eventually modifying its syntactic and nominative nature to become a name of a mythological character. As some examples, let us consider names that have emerged from the text of curses or folk songs.

*Case 1.* In Russian, there is a construction *Дай Бог* (*Dai Bog*), 'God grant, God willing'. Originally, this is an appeal to God, which has the meaning 'Let it be so'. This construction can also take the opposite meaning of wishing evil on someone, as a curse (especially in the inversive form *Bog dai*, 'God send'): *Бог дай тебе провалиться!* lit. 'God let you fall through the ground!' ('Hell with you!'). Since curses are appeals to the devil rather than to God, the formula *Bog dai* appears to be encoded within the phrase. This formula undergoes syntactic contraction, morphological transposition, and phonetic contraction: *Бодáй те провалиться*, lit. '*Bodái* you fall' (Yenisei dialect); *Богдай тебя* (*тебе*), lit. '*Bogdai* (to) you' (Smilensk, Tambovsk, Voronezh dialects); *Бодай тебе*, lit. '*Bodai* to you' (Voronezh, Smolensk, Tambovsk, Yenisei dialects) ([Filin et al. 1965](#), vol. 3, pp. 47, 54); etc.[4] At the same time, there is a secondary connection to the imperative form of the verb *бодать* (*bodat'*), 'to stab with horns', which fits well semantically into the range of words that are predicates of ill-wishes. This is particularly notable in curses like *Хвороба тебя бодáй*, 'May illness stab you with horns' (Krasnoyarsk dialect) ([Fel'de 2003–2010](#), vol. 5, p. 56), etc. At the next stage of transformation, the imperative form *бодай* becomes a noun, a subject: *Забери тебя бодáй*, 'May *bodái* take you' (Volgograd dialect) ([Mokienko and Nikitina 2013](#), p. 49). Thus, here emerges a certain creature capable of independent action. This creature can "leave" phrases of ill-wishes and move to another textual genre—threats. This is accompanied by further de-etymologization; *бáдя* (*bádia*), *бадя́* (*badiá*), or *бадяй́*(*badiái*) is defined in dictionaries as a 'mythical creature used to scare children': *Раньше детей бадей пугали: "Бадя тебя унесет", а никто его не видел*, 'Adults used to scare children with *badia*: *Badia* will take you away!—but nobody ever saw him'; *Вот бадя придет и заберёт тебя*, '*Badia* will come and take you away'; *"Бадяй придёт, в сумку тебя посадит",—детей пугали, кто не слушался*, '*Badiai* will come and put you in his bag—adults scared children who didn't behave'; *Не реви, бадяй заберет*, 'Don't cry, *Badiai* will take you away' (Vologda dialect) ([Gerd 1994–2005](#), vol. 1, pp. 39–40, 28). Moreover, *badia* is found outside the established folklore formulas as *бáдя, бадя́* (*bádia, badiá*), meaning 'invalid, cripple': *А то бадей какого-то инвалида назовут, вот как бадя идёт*, 'They start calling some disabled person *badia*, like: Look, *badia* is walking!'; *Такой у нас был бадя безрукой и немой, ходит да говорит "мня-мня-мня". Как скажешь, что бадя идёт, так все ребята разбегутся*, 'We had a *badia* who was armless and mute, he was walking and saying "*mnia-mnia-mnia*". As soon as you say that *badia* is coming, all the children run away' (Vologda dialect) ([Ibid.](#), p. 39). *Badia* also enters the literature on mythology as a separate mythological character: cf. "*бадай, бадя, бадяй, бадяйка* (*badai, badia, badiai, badiaika*)—a bogeyman adults use to scare children. <…> The outward appearance of *badiai* is vague and mysterious. It is someone hideous, he is often numb and armless, and sometimes lame. He kidnaps children" ([Vlasova 2008](#), p. 28; the same: [Cherepanova 1996](#), p. 165).

In this case, the de-etymologization of the word reaches the highest degree, which is why the reputable etymological dictionary of Alexander E. Anikin provides the following commentary on the words *бáдя, бадя́*, and *бадяй* (*bádia, badiá, badiái*): "Unclear. Probably a borrowing" ([Anikin 2007](#), vol. 2, p. 48).

*Case 2.* Another representative case, also originating from curses, is the Arkhangelsk dialectal word *желвáк* (*zhelvák*), meaning 'a being belonging to the evil forces' (an example

is described in Berezovich and Surikova 2018, p. 104). Originally, *желвáк* in the Northern Russian dialects refers to an abscess, or boil. This word and its phonetic and word-formational versions are constantly used in curses and other kinds of expressive phrases, such as *Желвáк тебе в горло*, '(May you have) a *zhelvák* (boil) in your throat' (Olonetsk dialect), and *Желвáк тебе в рот*, '(May you have) a *zhelvák* (boil) in your mouth' (Perm dialect). Gradually, the meaning of the word *желвáк* in invective has become blurred and generalized to the very broad meaning of "malicious figure". This is how such phrases as *Понеси тебя желвак*, 'May *zhelvak* carry you' (Archangelsk dialect), arise. Here, *zhelvák* is perceived as a personified character and is functionally equivalent to a devil or a *leshy* (forest spirit). Based on such invectives, **zhelvák** creates its own mythology that goes beyond curses. This word appears in the narratives of speakers of the Arkhangelsk dialects. According to these narratives, a **zhelvak** can be found in specific locations, such as the Emetsk village, and can carry away someone who does not "shut away" from him, and so on.

*Case 3.* Another example (widely known among Slavic folklorists) refers not to the folk tradition, but to the speculative study of this tradition. In the works of some specialists on Slavic mythology and writers (such as Miechowita, Kromer, Stryjkowski, Giesel, Ostrovsky, Derzhavin, etc.), written in the 16th–19th centuries, there occurs the name *Лель* (*Lel'*), which is interpreted as the name of the deity of marriage and the sun, etc. *Lel'* is often presented together with other characters, such as *Полель* (*Polel'*) and *Лада* (*Lada*) (see Sumtsov 1881, pp. 46–48, etc.). Image of *Lel'* are also found in the literature. Among the best-known examples are Pushkin's poem *Ruslan and Lyudmila*, Mickiewicz's *Pan Tadeusz*,[5] and Alexander N. Ostrovsky's play *The Snow Maiden*. Most often, the word *лель* (*lel'*) and similar-sounding combinations appear in the refrains of Slavic wedding songs (for example, the Russian *Ой лелю, молодая, о лелю*, '*Oi leliu*, the young [bride], o *leliu'*). Some "armchair" linguists have claimed that *лелю* is a vocative form of *лель*, i.e., an address to a character of this name. A brilliant critical analysis of such instances was carried out by Alexander A. Potebnja, the great 19th century philologist. Potebnja refers to them as "scholarly fairy tales" (Potebnia 1883, p. 17) and points out that "our predecessors too hastily elevated <…> [the song words *lel*, *leliu*] to the proper name of a deity of marriage, the sun, etc." (Ibid., p. 20); "*Lel'*, as the name of a deity imposed by the 18th century tastelessness even to Pushkin <…>, is not a fact, but a very shaky guess" (Ibid., pp. 16–17). What we have here is not an appellation to a deity, but ordinary song refrains. As Nikita I. Tolstoy points out, the refrains *алё-ле*, *ай люли*, *люли-люли*, *лелею* (*alio-le*, *ai liuli*, *liuli-liuli*, *leleiu*), etc. originate from the cry *hallelujah*, used in church rites, coming from the Greek ἀλληλούια < Ancient Hebrew *hallᵉlū-jāh* "praise the Lord" (Tolstoy 1995, p. 100).

Although the example above is related to "armchair" mythology, there are also some analogies in folk tradition. For instance, working in the north-east of the Kostroma region, our expedition recorded the texts of ritual songs which used to be sung during the New Year period and were aimed at fortune-telling. One example of such a song is called *úлия* (*íliia*), and this word features in the refrain, most often in the form *úлию-úлию* (*íliiu-íliiu*) (DRND). This is a variant of the *лелею*-type refrains mentioned above; therefore, *úлия* (*íliia*) can also be traced back to the exclamation "Hallelujah!" In some texts, a character named **Илия/Илья** (**Iliia/Ilja**) is "singled out" from the refrain *úлию-úлию* and begins to act: *Ходит Илия по полю, Стога считает*, 'Iliia walks in the field, counts haystacks' (DRND). Apparently, the transformation of *íliia* into a character is also influenced by the name *Iliya/Ilja* 'Elijah', a well-known Old Testament saint.

## 5. Conclusions

The classification of cases presented in this paper makes no claim to comprehensiveness. We have described only a few cases out of a multitude. What is more important for us is that the cases described here are not random or exceptional, but systematic. The diversity of examples proves the scale of the phenomenon under study. This phenomenon refers to the creation of a new, previously non-existent mythological character based on a linguistic sign (most often onomastic) and the resources that already exist in the system of

collective ideas. It takes two motivational sources for a new character to emerge—the linguistic (a word that seeks a new context), and the cultural (a semiotically intense context: a situation associated with danger, prohibition, omen, aggression, or magical practices). These incentives are often combined in cultural linguistic practices, so the processes of "re-reproduction", the constant renewal of mythological material, and the living oscillation of fragments of the religious–mythological system are natural and expected. The linguistic sign therefore becomes both a cause and an indicator of these processes.

**Author Contributions:** Conceptualization, O.D.S. and E.L.B.; methodology, O.D.S.; software, E.L.B.; validation, E.L.B.; formal analysis, O.D.S.; investigation, O.D.S.; resources, E.L.B.; data curation, E.L.B.; writing—original draft preparation, O.D.S.; writing—review and editing, E.L.B.; visualization, E.L.B.; supervision, E.L.B.; project administration, E.L.B.; funding acquisition, E.L.B. All authors have read and agreed to the published version of the manuscript.

**Funding:** The research was funded by the Russian Science Foundation, grant number 23-18-00439 Onomasticon and Linguocultural History of European Russia, https://rscf.ru/en/project/23-18-00439 (accessed on 7 October 2023).

**Institutional Review Board Statement:** Not applicable.

**Informed Consent Statement:** Not applicable.

**Data Availability Statement:** Data available in a publicly accessible repository that does not issue DOIs. Publicly available datasets were analyzed in this study. This data can be found in the *Card Index of the Toponymic Expedition of Ural Federal University* (Yekaterinburg: Ural Federal University).

**Acknowledgments:** The authors would like to thank Vladimir V. Napol'skikh for his valuable advice and interesting examples.

**Conflicts of Interest:** The funding sponsors had no role in the design of the study; in the collection, analyses, or interpretation of data; in the writing of the manuscript; and in the decision to publish the results. The author declares no conflict of interest.

## Notes

[1] The brook name *Voeshnyi* is not related to the Russian verb *vyt'*, 'to howl', but is of a substrate (Finno-Ugric) origin (Matveev 2001, p. 259).

[2] As opposed to folk etymology, academic etymology claims that the name of Mount Azov comes from the Bashkir–Tatar word *azau*, meaning 'molar tooth': 'on the top of the mountain, there is a rock resembling a molar tooth' (Matveev 2008, p. 10).

[3] A female mythological character similar in cultural motivation is known among the South Slavs. Bulgarian and Macedonian ballads record ideas about a mythological creature named ***Sviataia Nedelia***, meaning 'Holy Sunday'. She is covered in blood, her body is stabbed, and her clothes are torn. These features reflect the consequences of violating the prohibition to sew, cut, etc. on Sundays (see Sedakova 2008).

[4] Cf. Belorusian *Багдай ты акалеў*, 'May you die' (Grynblat 1979, p. 202).

[5] Cf. *Polish Kastor z bratem Polluksem jaśnieli na czele, Zwani niegdyś u Sławian Lele i Polele*, 'Castor and his brother Pollux glittered at their head, once called among the Slavs Lele and Polele' (A. Mickiewicz).

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
