# Peer review of "From Name to Myth (Based on Russian Cultural and Literary Tradition)"

_religions, doi:10.3390/rel14111412_

Round 1
Reviewer 1 Report
Comments and Suggestions for Authors
The authors present a classification system for the origins of Russian myths, which they state may be further developed. This in itself is a significant contribution to the existing scholarship. They take a largely semiotic approach, which reveals linguistic insights. In addition, they have been doing case studies for a long time, and they present compelling examples to support their classification system.
The one suggestion I could make is that the authors provide more context on the existing research on this topic; however, I do not think it is absolutely necessary.
Comments on the Quality of English LanguageThe English is not bad, but I think it would be helpful to have a native speaker in academia proofread the article for style.
Author Response
Comment 1: The one suggestion I could make is that the authors provide more context on the existing research on this topic; however, I do not think it is absolutely necessary.
Response 1: Thank you very much for taking the time to review this manuscript. As you note the unnecessarity of correction, we feel ourselves allowed to leave the things as they are.
Comment 2: The English is not bad, but I think it would be helpful to have a native speaker in academia proofread the article for style.
Response 2: Sure, it would be better, but unfortunately we have not this opportunity.
Reviewer 2 Report
Comments and Suggestions for Authors
The article based on the material of Russian dialects (dictionaries of folk dialects and field materials of the Ural Federal University - Russian North, Ural and Volga regions). It analyzes the occurrence of unique/rare mythological characters based on two types of linguistic signs: 1) normative vocabulary (toponyms, chrononyms, etc.), 2) names from (folklore) texts. The article argues that everyday mythology is a developing and evolving system.
The authors come to the conclusion that for the appearance of a new mythological character, two motivational sources are necessary - linguistic (name) and cultural (situation associated with danger, prohibition, omen, aggression or magical practices).
Some notes:
495-496: Position of paragraph?
497-498: There are no references to Driver et al. 2000 in the text.
Many thanks to authors for very interesting article!
Author Response
Comments 1: 495-496: Position of paragraph?
Response 1: Fixed.
Comments 2: There are no references to Driver et al. 2000 in the text.
Response 2: The link is removed.
Comments 3: Many thanks to authors for very interesting article!
Response 3: Thank you very much for taking the time to review this manuscript!
Reviewer 3 Report
Comments and Suggestions for Authors
Just some affirmations, concepts or expressions need to be clarified by the authors (see revised paper in the attachment).

Author Response
Comments 1: Just some affirmations, concepts or expressions need to be clarified by the authors.
Response 1: Thank you very much for taking the time to review this manuscript. All bugs have fixed. The short cover letter detailig the changes is attached.
